

# ET Cool Home: Innovative Educational Activities on Evapotranspiration and Urban Heat

Kyle Blount[1], Garett Pignotti[2], Jordyn Wolfand[3]

[1]Environmental Studies, University of Illinois Springfield, Springfield, IL 62704, USA
[2]ORISE Fellow, US Forest Service Pacific Northwest Research Station, Corvallis, OR 97331, USA
[3]Civil & Environmental Engineering, University of Portland, Portland, OR 97203, USA

*Correspondence to*: Kyle Blount (wblou2@uis.edu)

**Abstract.** Teaching evapotranspiration (ET) in university courses often focuses on either oversimplified process descriptions or complex empirical calculations, both of which lack grounding in students' real-world experiences and prior knowledge.
This calls for a more applied approach to teaching about ET that connects concepts to experience for improved educational outcomes. One such opportunity exists at the intersections between ET and heat in cities, where a growing majority of the world's population lives, including many of our students. In this work we describe an ET educational activity that integrates theory with practical design, taking advantage of the close link between ET processes and urban heat patterns. In a benchtop experiment, students measure ET variations across common land surfaces (e.g., asphalt, grass, and mulch) through water and
energy balance approaches. The experiment is paired with an "urban heat tour" in the campus environment, facilitated by portable infrared cameras, offering firsthand observation of urban heat patterns. These two activities, together, provide context in which students can understand the difference in ET across various land covers, describe the relationship between ET and land surface temperatures, and explain the impacts of urban design on heat dynamics. The activities are adaptable to serve a diversity of student backgrounds and to different educational contexts, including public demonstrations and K-12 classrooms.

**Short Summary.** We introduce an applied approach to evapotranspiration (ET) instruction. In a laboratory experiment, students calculate ET using both water balance and energy balance approaches for five representative urban land surface covers (gravel, soil, grass, asphalt, and mulch). The experiment is paired with an urban heat tour facilitated by thermal infrared cameras. The activities are adaptable for various contexts, ranging from undergraduate lab classes to K-12 demonstrations.

## 1 Introduction and Motivation

Evapotranspiration (ET) is frequently covered in undergraduate curricula spanning geology, earth and environmental science, environmental and water resources engineering, and physical geography courses. Lower-division courses often introduce ET as the general process involving the movement of water from the land to the atmosphere through evaporation and transpiration. In upper-division courses in hydrology and water resources engineering, ET instruction centers on highly
theoretical approaches and applications of complex equations, such as the Penman-Monteith approach. However, these





approaches often lack scaffolding, presenting an opportunity for an improved, more tangible teaching approach to ET in both lower- and upper-division courses.

ET is a key component of the hydrologic cycle, and a quantitative evaluation of ET is essential for understanding the impact of climate change on hydrology, water resource management, ecosystem management, and food security. Critically,
ET is a link between water and energy in the environment. This link, particularly relevant to highly heterogeneous, anthropogenic land covers in urban areas, has significant implications in determining heat fluxes. Alterations to the land surface in these regions affect not only ET but also the surface energy balance, which can intensify urban heat islands. However, despite their connection, the topics of ET and urban heat are often loosely linked or taught in entirely separate courses.

In urban regions, ET provides students an opportunity to use a practical framework for observing ET through both
water and energy balances with societally relevant implications for environmental and urban management. In response to these challenges and opportunities, this paper presents an educational activity that applies water and energy balance approaches to demonstrate and measure ET across urban land covers and meaningfully engage with management implications. This activity is adaptable in complexity for a variety of contexts, ranging from a qualitative demonstration of ET and heat differences across land covers to a more advanced, quantitative activity where students calculate and compare water and energy balance ET
estimates across land covers.

## 2 Theoretical Basis

### 2.1 Evapotranspiration: Theory and Measurement

ET is defined as the sum of all water movement from the land surface to the atmosphere from a combination of evaporation (primarily from soils, open water, and vegetation), sublimation of ice and snow, and transpiration from plants
(Hagan et al., 1967; Dingman, 2015). Approximately 62% of global continental precipitation becomes ET, with 97% occurring over land surfaces and 3% over open water (Dingman, 2015).

Quantifying ET, however, necessitates precise knowledge of each process within a study area at high spatiotemporal resolution across heterogeneous land surfaces. In urban areas, water is often the limiting factor controlling ET rates, especially for impervious surfaces that store less water than pervious soils (Jongen et al., 2022). Resolving ET fluxes across space and
time requires high-quality data that is more difficult and expensive to measure than for other components of the hydrologic cycle. Therefore, direct ET measurement is often impractical, leading to approximations of ET flux using empirical equations of potential ET like the Penman-Monteith, Priestley-Taylor, or Hargreaves Samani, which use more readily available climate data such as solar radiation, humidity, and wind speed as inputs (Priestley & Taylor, 1972; Hargreaves & Samani, 1985; Allen et al., 1998; Glenn et al., 2007; Dingman, 2015). Two common approaches to estimating ET, the water and energy balance
methods (Fig. 1), are described below.



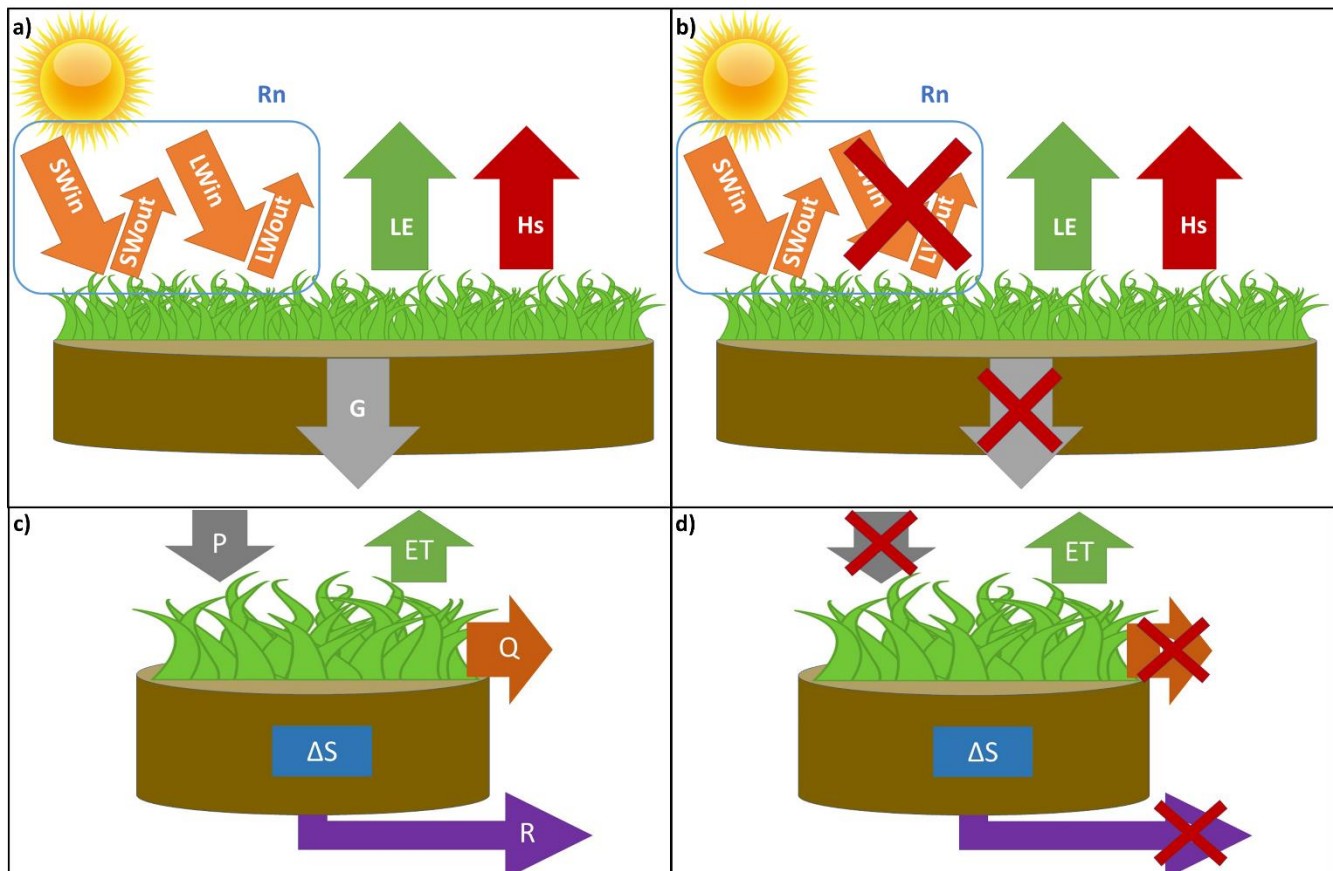

**Figure 1. Energy balance and water balance system diagrams used in the classroom activities showing the (a) complete energy balance, (b) simplified energy balance for the current activity, (c) complete water balance, and (d) simplified water balance used for the current activity.**

### 2.1.1 Water Balance Methods

These approaches apply a water balance equation (Eq. 1) to a defined system, such as a lake, watershed, or soil column. By measuring the other components of the hydrologic cycle – including precipitation (P), runoff (Q), changes to soil moisture (ΔSM), and recharge (R) – ET can be calculated as the remaining unknown in the water balance equation (Villagra et al., 1995; Zhang et al., 2002; Dingman 2015). The water balance method is often used with weighing lysimeters, highly instrumented underground devices filled with soil and vegetation on a scale, by measuring change in water weight (Yang et al., 2000; Loos et al., 2007). In this case, the water balance method is also referred to as a mass balance method because changes to the mass of water within all components of the system are directly measured.

$$ET = P - Q - R - \Delta SM \qquad (1)$$





### 2.1.2 Energy Balance Methods


ET is approximated using an energy balance approach by quantifying the energy used during water evaporation, known as latent heat flux (LE). Because the latent heat of vaporization of water represents a predictable amount of energy used to evaporate a known quantity of water, LE is readily converted into ET; in this activity, we assume the small temperature dependencies of LE are negligible for simplicity (De Bruin et al., 1982; Shuttleworth, 2008; Dingman, 2015). The energy

balance approach averages radiative energy transfer at the land surface – including net radiation ($R_n$) comprised of incoming and reflected long- and short-wave radiation (LW and SW, respectively, Eq. 2), ground heat flux (G), and sensible heat flux ($H_s$) – over a known period to calculate the energy used to evaporate water as the remaining unknown in Eq. 3 (Fritschen, 1965; Dingman, 2015).

$$R_n = (SW_{in} + LW_{in}) - (SW_{out} + LW_{out}) \tag{2}$$

$$LE = R_n - H_s - G \tag{3}$$

These two categories of approaches allow ET calculation across various spatiotemporal scales with advantages over direct observations of ET from eddy covariance (EC) towers including: (a) EC measurements assume a smooth, horizontal surface relative to measurement height, which are inconsistent in highly heterogeneous urban environments at all but neighborhood scales and (b) even at scales where EC measurements can be applied, land cover heterogeneity prevents

identification of contributions of ET from various sources (Coutts et al., 2007; Foken et al., 2011). Energy and water balance approaches are also more cost effective than EC and therefore represent two of the more commonly applied methods in hydrology for ET estimation (Shuttleworth, 2008; Dingman, 2015).

### 2.2 Urban Heat

Cities are unique environments primarily due to the extensive presence of human society and the direct, anthropogenic

manipulation of the environment. Their traits stem largely from the alteration of the urban land surface, including removing vegetation and covering soils with impervious surfaces. Consequently, hydrology in urban environments is altered leading to reduced infiltration and ET coupled with increased runoff (Oswald et al., 2023). These changes also affect the surface energy balance, impacting how surfaces reflect, store, or release heat, which increases sensible heat fluxes. This results in city air temperatures of up to 8°C hotter than surrounding rural areas, and these increases may be even more extreme during extreme

heat events (National Integrated Heat Health Information System, 2023; Chen et al., 2023). Adding complexity to these trends, cities are highly heterogeneous, which causes temperature to vary greatly in space and time based on local land cover. These observations motivate questions of where heat is occurring, its severity, and why it is occurring there, all of which relate to land cover characteristics and their influence on interactions between water and heat.

Changes to the urban energy balance have impacts on the environment, especially during heat waves. The altered

biophysical function induces stress and mortality in trees, increases energy consumption, and negatively impacts human health



(Heaviside et al., 2017; Li et al., 2019; Tong et al., 2021; Yadav et al., 2023). Notably, extreme heat is the leading cause of weather-related deaths in the United States and contributes to approximately 489,000 deaths annually worldwide (Zhao et al., 2021; National Weather Service, 2022). Exacerbation of these issues is expected, driven by anthropogenic climate change, that will result in hotter cities with more frequent heat waves. Urban populations now represent a growing majority of the world's population, increasing the number of people affected by urban heat (United Nations, 2017). These impacts are also unequally distributed, with low-income and primarily minority communities in the United States experiencing higher temperatures associated with less greenspace and canopy cover (Mitchell & Chakraborty, 2018; Hsu et al., 2021). Driven by historic disinvestment, these areas are less likely to have adequate resources to mitigate the impacts of heat waves, such as air conditioning (Romitti et al., 2018). Heat mitigation strategies are, therefore, increasingly important for equitable development in a changing climate.

Communicating these core concepts in urban hydrology, microclimate, and biophysics is critical within highly heterogeneous urban landscapes given the societal implications arising from their unique characteristics and associated risks. Although climate change and associated extreme heat are global issues, local governments are predominantly directing decisions related to development and redevelopment, zoning, and nature-based policies. This often results in localized and fractured governance of urban heat (Keith et al., 2023). Furthermore, the thermal characteristics of individual properties, including built area and the extent and type of vegetation and canopy, are strongly influenced by the decisions of property owners. It is therefore key to cultivate locally informed community members and professionals, including our students, who can advocate for and promote effective heat-management strategies.

## 2.3 Connecting ET and Urban Heat

The interactions between water and energy directly link the hydrologic cycle, and ET in particular, to the formation of urban heat islands. Net radiation is apportioned into G, LE, and $H_s$, but because G is small over long periods of time (approximately two orders of magnitude smaller than $R_n$), we can assume that it is equal to zero (Trenberth et al., 2009). Equation 3 then illustrates that net radiation is partitioned between either latent heat flux (ET) or sensible heat flux (increased temperatures). This means that energy that is absorbed by the land surface is partitioned to either evaporate water or increase temperatures, regulated by the land surface properties and available moisture. In urban areas, the proliferation of impervious and non-vegetated surfaces reduces moisture availability, decreasing latent heat flux while increasing sensible heat flux. Although other characteristics of urban landscapes such as thermal properties of impervious surfaces, geometries, and shading influence these dynamics, the land cover change and the associated shift in $LE/H_s$ partitioning fundamentally governs the generation of urban heat islands (Theeuwes et al., 2014; Loridan & Grimmond, 2012; Gülten et al., 2016; Mohajerani et al., 2017).

The educational activity described herein leverages the interaction between water and energy to ground ET estimation approaches to the societally relevant issue of urban heat islands. The activity consists of a benchtop land surface experiment designed for calculating ET using both water balance and energy balance approaches employing a factorial design that accounts

for differences in urban land cover and surface moisture content. Additionally, the activity includes a guided urban heat tour

that allows direct observation and discussion of urban heat. The application of this activity provides students opportunities to think critically about urban environmental management and the choices and trade-offs associated with urban design, city development, and heat and water dynamics. For less quantitative classroom iterations or general public demonstrations, the activity allows participants to interact with and reflect on differences in their urban environments and personal experiences developing connections between the decisions they make in their environment and resulting biophysical outcomes.

## 3 Learning Objectives and Overview


The overarching goal of the activity is to expose students to the relationship between ET and environmental management considerations, specifically focusing on the connection between urban heat and land surface cover. At the end of the activity, students should be able to: (1) *apply* the energy balance equation to determine ET rates; (2) *explain* how ET rates vary over different land surface covers; and (3) *describe* how ET and land surface temperature relate and explore how this may

impact urban design. The activity has two main components: a land surface experiment and a campus heat tour.

## 4 Land Surface Experiment

### 4.1 Materials and Set-up

Ten land surface cores, representing five common urban land covers (grass, bare soil, gravel, asphalt, and mulch), were constructed and positioned under a mounted heat lamp (Fig. 2). While the land surface cores were constructed in the lab,

they could alternatively be collected from the urban environment. The cores were cylindrical, and constructed from 4.7 cm diameter tubing, cut to approximately 4 cm tall and capped along the bottom. Half of the land surface cores (one per surface type) were wet with approximately 40 mL of water (to the top of the surface material inside the tube) to represent wet conditions. Students performed initial observations and measurements, recording time, initial mass, and initial surface temperature. Analytical balances were used to record the mass of each core, while infrared thermometers were used for

temperature measurements. Students also measured the diameter of the cores and calculated the corresponding surface area. The cores were then positioned beneath a 250 W heat lamp, and a FLIR T650sc thermal camera, mounted on a tripod, allowed students to visually observe differences in surface temperature. Students took final surface temperature and mass measurements approximately 90-120 minutes after their initial measurements.







**Figure 2. Land surface experiment set up with 15 cores (three per land cover type), heat lamp, and infrared camera.**

## 4.2 Water Balance

Because the land surface cores are functionally weighing lysimeters that do not experience precipitation, deep drainage, or runoff during the experiment, ET is equal to the change in water storage within the core (Fig. 1c-d). Students, therefore, calculated ET rate (mm d$^{-1}$) using a simplified water balance approach (Eq. 4), where $M_i$ and $M_f$ are initial and final masses of each core, $\rho_w$ is the density of water, A is core surface area, and $\Delta t$ is time.

$$ET_{WB} = \frac{M_i - M_f}{\rho_w * \Delta t * A} \tag{4}$$







## 4.2 Energy Balance

Students then calculated ET (mm d$^{-1}$) using an energy balance approach (Eq. 5), where LE is latent heat flux (W m$^{-2}$), l$_v$ is the latent heat of vaporization for water, and $\rho_{H_2O}$ is the density of water.

$$ET_{EB} = \frac{LE}{l_v * \rho_{H_2O}} \tag{5}$$

Given that both long wave radiation and ground heat flux are typically 1-10% of net radiation and often assumed negligible (Trenberth et al., 2009), latent heat flux becomes a function of incoming short-wave radiation (SW$_{in}$), reflected short-wave radiation (SW$_{out}$), and sensible heat flux (H$_s$) (all units of W m$^{-2}$), as shown in Fig. 1b and Eq. 6.

$$LE = SW_{in} - SW_{out} - H_s \tag{6}$$

By substituting outgoing short-wave radiation as the product of albedo (α) and incoming short-wave radiation (Eq. 7), Eq. 6 can be rearranged to solve for latent heat flux (Eq. 8).

$$\alpha = \frac{SW_{out}}{SW_{in}} \tag{7}$$

$$LE = SW_{in}(1 - \alpha) - H_s \tag{8}$$

Sensible heat flux can be estimated with Eq. 9, where $\rho_{air}$ is the density of air, $C_{\rho\_air}$ is the specific heat capacity of air, T$_s$ is surface temperature, T$_a$ is air temperature, and r$_a$ is aerodynamic resistance.

$$H_s = \rho_{air} * C_{\rho\_air} * \frac{T_s - T_a}{r_a} \tag{9}$$

## 5 Urban Heat Tour

While the land cover cores were heating, students were taken on a "campus heat tour." Organized into groups of 2-3, students were given thermal infrared camera attachments for their cell phones (FLIR ONE®, Teldyne FLIR, Wilsonville, OR). They were then instructed to explore campus through the cameras and make observations, noting anthropogenic heat sources such as buildings and vehicles. Students measured surface temperatures of urban land covers that corresponded to those simulated in the laboratory activity, including mulch, asphalt, and grass while also considering the presence of vegetation and other shading. Subsequently, the class engaged in a brief discussion to facilitate critical thinking about the connection between land surface properties and heat. Example discussion questions include:

- How does land cover impact land surface temperature?
- Can you identify specific practices that contribute to increased temperatures in certain parts of campus?
- What urban planning and design strategies could be implemented to mitigate urban heat?



## 6 Example Implementation

### 6.1 Application

The lab activity was implemented at three universities – University of Portland, University of Washington Vancouver, and University of Illinois Springfield – with undergraduate students primarily majoring in civil engineering or environmental science in their third and fourth year. The lab was conducted during a 3-hour lab course block, requiring approximately 2-4 additional hours of outside of class time to complete the lab assignment write-up.

### 6.2 Sample Results

Assumed constants (at 20°C and 1 atm) and incoming shortwave radiation, measured using a METER PYR sensor (Meter Environment, Pullman, WA), were provided to all students (Table 1).

**Table 1. Assumed constants for student calculations**

| Constant Description | Symbol | Value | Units |
|---|---|---|---|
| Density of Water at 20°C and 1 atm | $\rho_{H20}$ | 0.99823 | g cm$^{-3}$ |
| Density of Air at 20°C and 1 atm | $\rho_{air}$ | 1204.7 | g m$^{-3}$ |
| Specific heat capacity of air at 20°C and 1 atm | $Cp_{air}$ | 1006.1 | J kg$^{-1}$ K$^{-1}$ |
| Air Temperature | $T_a$ | 293.15 | K |
| Aerodynamic resistance | $r_a$ | 208 | s m$^{-1}$ |
| Incoming shortwave radiation | $SW_{in}$ | 348.1 | W m$^{-2}$ |
| Latent heat of vaporization for water | $l_v$ | 2.26 x 10$^6$ | J kg$^{-1}$ |

In addition, typical albedo values for each surface type were provided to all students based on published values (Table 2), with distinctions provided for wet and dry conditions for surfaces where moisture content alters albedo by more than 0.1 (Kotak et al., 2015; An et al., 2017; Chen et al., 2019; Qin et al., 2023).

All data were recorded in units of grams (mass), Kelvin (temperature), square centimetres (area), and minutes (time). Conversion factors and unit conversions used to complete ET calculations are provided in the Supplemental Material.





**Table 2. Assumed albedo values for each land surface from literature**

| Surface | Albedo ($\alpha$) |
| --- | --- |
| Asphalt | 0.05 |
| Wet gravel | 0.3 |
| Dry gravel | 0.45 |
| Wet mulch | 0.20 |
| Dry mulch | 0.25 |
| Wet soil | 0.15 |
| Dry soil | 0.20 |
| Grass | 0.26 |

220

    Student-estimated ET rates ranged from 0-10 mm/d using the energy balance method to 0-16 mm/d using the water balance approach (Fig. 3). Our discussion of these sample results is framed with four discussion questions provided to the students. Note that potential answers will vary, depending on the students' results as well as their interpretation of the results.



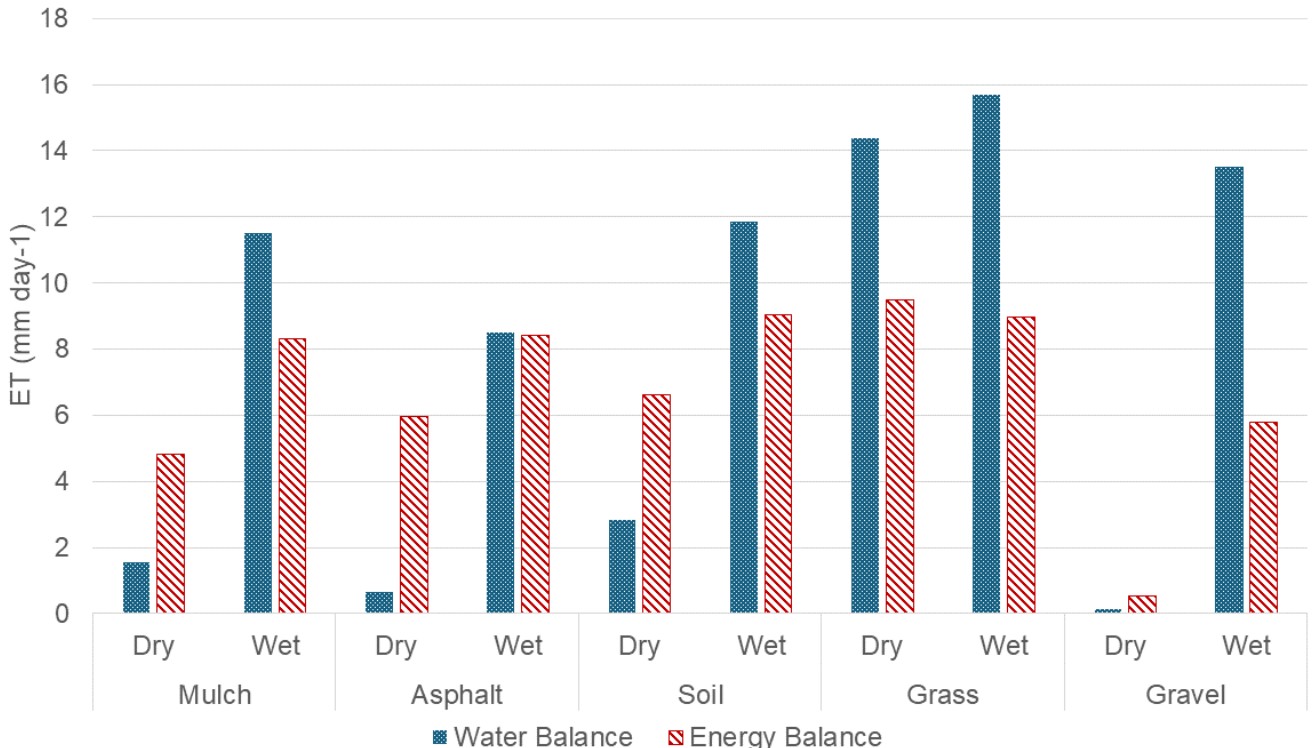

**Figure 3. Student sample results using the energy balance method (red, striped bars) and water balance method (blue, dotted bars).**

1. Briefly discuss your results. How do evapotranspiration rates across different land surfaces compare? Is there a difference when a land surface is wet or dry? Is there a difference when estimated with the mass vs. energy balance approach?

    a. ET rates varied across different land surfaces from nearly zero (dry gravel) to about 16 mm/d (wet grass). Clear differences in ET were observed between wet and dry surfaces, particularly for anthropogenic land surfaces with limited moisture retention in urban environments. For example, the dry asphalt, gravel, and mulch showed negligible mass differences after 100 minutes. ET rates were highest for grass, due to its transpiration potential. Considerable differences emerged between ET estimates from the mass balance versus energy balance approach. The water balance approach produced higher ET rates for all wet land surfaces and dry grass, while the energy balance approach resulted in higher ET rates for dry mulch, asphalt, soil, and gravel. These difference may relate to guiding assumptions made within the study design.

2. How does replacing pervious surfaces (e.g., grass and soils) with impervious surfaces (e.g., mulch, gravel, and asphalt) change latent heat fluxes (or ET)? Sensible heat fluxes (heat)?





a. Replacing pervious surfaces with impervious surfaces reduces potential ET, thereby reducing latent heat flux. Sensible heat, felt from the land surface, increases as pervious surfaces are replaced with impervious surfaces.

245

3. How do impervious surfaces affect heat in cities? What are the possible impacts of this on people and the environment?

a. Impervious surfaces contribute to increased heat in cities. This can negatively impact human health by causing heat-related illnesses such as heat stroke and exacerbate existing conditions such as asthma. Increased heat can also lead to secondary effects, including the increased demand for resources such as energy (for cooling) and water. Excess heat can additionally disrupt habitats, particularly affecting temperature sensitive aquatic species. These impacts are compounded by increased heat due to climate change.

255

4. As the climate changes and temperatures rise, urban heat islands pose greater threats to cities worldwide. If you were a city manager, propose how you could use these results to lessen the impact of heat in cities. Are there any potential downsides or trade-offs to your recommendations?

a. Mitigating urban heat requires designing cities to be cool. For example, using materials with a high albedo can reflect more incoming solar radiation, while increasing pervious areas allows for moisture storage, increasing ET and cooling. Additionally, cities can plant more trees to increase shading. Downsides to these recommendations include potential increased costs and an increased demand for irrigation (and therefore water) to support vegetated areas.

## 7 Discussion

### 7.1 Placing the results into context

265     The laboratory activity's core learning objectives are reinforced by students connecting experimental-scale observations to real-world urban heat issues. In a successful implementation, students will directly link land surface properties to their role in amplifying or abating heat. This process, therefore, relies on students deducing differences in temperature across the test surfaces as a product of land surface characteristics, including water content. Students can then draw conclusions as to how changes in land surface properties modify the water and energy balance and its relation to heat regulation.

270     As students identify how different surface properties affect heat, it establishes a touchstone from which they can interrogate urban management strategies. The campus tour reinforces this connection through discussion on commonly observed landscape compositions and materials in an urban environment. By applying the principles of heat adsorption and





ET cooling, students can envision how different compositions of land surfaces may scale for highly urbanized areas in contrast to rural landscapes (i.e., the urban heat island concept). This should allow students to directly describe which specific properties promote warmer temperatures, such as high impermeability and low reflectivity.

More in depth discussion should encourage students to apply their understanding of urban heat islands and their drivers to propose management approaches that mitigate heat. This could focus on a range of management approaches, including increasing albedo of surfaces or promoting infiltration using permeable pavements – features possibly observed during campus heat tours. Students must then consider trade-offs in cooling with practical urban design. In our discussions, for example, we found a common suggestion was to increase the amount of water applied to encourage ET cooling effects. In practice, however, this may strain already limited water resources (Gober et al., 2009; Cuthbert et al., 2022) and risk increasing humid-heat conditions (Zhang et al., 2023). Contextual discussion, tailored to area-specific hydroclimate and geography, is therefore warranted.

Highlighting known adverse impacts of urban heat (e.g., heatstroke, increased energy consumption for cooling, vegetative stress) helps ground the discussion in relatable experience and provides perspective. Of particular importance is the discussion of inequitable exposure to urban heat prevalent in most major U.S. cities (Hsu et al., 2021). A result of historic discriminatory practices (Nardone et al., 2021), urban heat inequities are a direct consequence of disinvestment in specific neighborhoods, associated with less greenspace, and therefore, cooling (Hoffman et al., 2020; Wong et al., 2021). Thus, these inequities can provide a powerful lens to stress the interdisciplinary nature of real-world challenges while underscoring the essential role primary scientific principles can play in their solutions.

## 7.2 Challenges and potential limitations

We conducted the educational activity at three separate universities, each with unique laboratory environments and diverse student groups. This provided an opportunity to evaluate and assess potential difficulties and constraints in lab execution within dynamic settings. Generally, the challenges identified fell into four categories: 1) resource limitations, 2) student background knowledge, 3) measurement and instrument errors, and 4) assumptions and generalizations.

We considered possible financial resource limitations a primary and acute barrier in developing our activity for general use. Consequently, we explicitly designed the activity to avoid high costs assuming access to typical laboratory equipment (e.g., analytical balances); however, we acknowledge that not all potential instructors will have full access to the necessary materials and equipment. A simple laboratory experiment set up, consisting of land cover cores, a heat lamp and bulb, and an infrared thermometer costs approximately $100 USD (~90 €) based on our purchase prices. This minimal set up allows for an energy balance ET calculation, and with additional access to standard weighing scales, water balance calculations are also possible. Activities with a thermal infrared camera can greatly increase costs. In our set up, thermal infrared cameras were borrowed from other laboratories or already available.





Teaching the lab for variable class sizes (between ~5-40 students), we found it best to place larger classes into pairs

or groups and assign responsibility for one set of land cover cores (i.e., one group is responsible wet and dry grass). This facilitated a coordinated approach where all students contributed cooperatively to a class data sheet, minimizing downtime between measurements and optimizing overall laboratory logistics. For classes much larger than 30 students, it may become necessary to have replicate cores and multiple infrared thermometers. It is important to note that because the lab requires time for ET to occur between measurements, we recommend 90 to 120 minutes to observe differences in water content and surface

temperature. We further recommend creating land surface cores at least one day before the experiment and setting up the experiment 15 minutes prior to the start of a laboratory period, including wetting and heating of the cores. Under time restrictions of less than an hour, we would expect the laboratory experiment may not yield desirable results (i.e., small or negligible changes in measured ET). In such circumstances, set up and initial measurements could be made prior to class, instead only requiring the second measurement during class time.

Specific laboratory design will need to fit a wide range of student background knowledge. While course-specific adjustments may emphasize or deemphasize theoretical aspects, we found it valuable to spend time introducing or reviewing key energy and water balance terms and their estimation (Fig. 1). We additionally found that many students had difficulty understanding differences between latent and sensible heat fluxes that required in-class time to explain these concepts. Similarly, regardless of mathematical background knowledge, we found students struggled to calculate ET using the energy

and water balance approaches, particularly with dimensional analysis and unit conversions (see Supplemental Materials). We therefore recommend demonstrating example calculations using these approaches. This is especially important in promoting an inclusive laboratory experiment that does not unintentionally exclude students uncomfortable with mathematics.

Measurement and instrument errors introduced two known spuriously high ET estimates in certain experiment iterations from: 1) a student recording error and 2) a water leak in one land surface core (gravel). Both errors led to unrealistic

ET calculations that required some level of intervention in facilitating student analysis. For the latter, we used the leak as an unintended example of deep drainage in a system, a key process we remove from our calculations, and as an opportunity for critical thinking. We also used several other assumptions in our calculations (e.g., albedo, ground heat flux, and long wave radiation values) that inexorably increase uncertainty. It is important to further discuss with students that these measurements are not completely representative of real-world land covers as we ignore variations in climatic factors (e.g., wind, surface

roughness, humidity) and diurnal and seasonal influence that affect ET rates. In our laboratory activities, we used these errors, assumptions, and uncertainties as learning opportunities to discuss broad challenges in scientific experimentation; however, we recognize eliminating this uncertainty may be more desirable. We expect that both careful construction of land surface cores and taking replicate measurements would help reduce errors.



### 7.3 Adaptations for different classrooms and general audiences

A key aspect of our activity is scalability for diverse participants, and it is easily adapted to meet various learning objectives and contexts. The activity was initially conceived for public outreach at a science museum and then reconfigured for the college level. The example implementation provided herein was designed for an undergraduate course in natural or environmental science or civil/environmental engineering with an emphasis on quantitative learning to supplement theory. For courses with less quantitative emphasis, providing students with spreadsheets or another tool for automatic calculations is

possible (see Supplementary Materials). Conversely, in more advanced courses (e.g., graduate level), students could be asked to pose their own questions to investigate, use scientific programming and tools for data analysis and visualization, and place their findings in the context of broader scientific literature. In situations without lab time or access, the activity could be used as a lecture aid, where the instructor performs measurements and calculations over a single or multiple class sessions.

    When interacting with the general public, lengthy experiments and calculations are impractical for learning. Instead,
effective demonstrations focus on visually engaging elements, interactivity, and relating concepts to common experiences (Reynolds 2009; Varner 2014). In our public demonstrations at the Oregon Museum of Science and Industry (OMSI) "Meet a Scientist" events (Fig. 4), the primary change was shifting focus from ET calculation to visualization. The land surface cores were set up similarly to the college-level experiment, with a mounted tripod so museumgoers could easily observe land surface temperatures. To create an engaging visual, we invited participants to spray ethanol onto the land surface cores and see the

rapid decrease in temperature in the thermal imagery associated with evaporative cooling, followed by an increase in temperature as the ethanol evaporates. While we also used this visualization in the classroom, the public demonstration relied on its interactivity and ability to distil a theoretical concept into a straightforward observation. Applying core science communication principles (Baram-Tsabari and Lewenstein, 2017; Cooke et al., 2017; Borowiec 2023), we asked participants to take on the role of investigator and determine which cores are hotter and why. To do so, we used the 2021 Pacific Northwest

heatwave (Schumacher et al., 2022; White et al., 2023) as an entry point to the material and relate it back to common experience. We then asked leading questions, careful to avoid any jargon, to prompt lines of investigation using the infrared thermometer and ethanol spray as tools to observe how ET can cool different surfaces. Our questions attempted to also ground high-level concepts in everyday experience. For example, asking which type of surface the participant would want to stand on during a hot day and why.






**Figure 4. Activity set up and demonstration modified for a general public audience. Presented during Oregon Museum of Science and Industry (OMSI) "Meet a Scientist" event on 7 May 2022.**

365       Because interest, background knowledge, and contact time highly varied, flexibility in tailoring the public activity to individuals was key. This includes providing an immediately engaging aspect. For young children, we used an inexpensive thermal mousepad that changes color when touched (e.g., https://grifiti.com/products/grifiti-mood-mouse-pad) and largely emphasized interactive and hands-on aspects of the activity, namely the ethanol spray and measuring temperature with the thermal infrared thermometer. This allowed us to discuss basic heat concepts in simple language. For adults, we asked more

direct questions to prompt critical thinking about how ET functions to cool the environment. We used sweating in humans as an analogy to better communicate this process. For interested participants, we asked them to extend these concepts further into how urban compositions may amplify heat and the consequent design implications. Including discussion of personal research related to the concepts in the demonstration was additionally helpful in communicating their broader significance.

For both the classroom and public outreach version of the land surface experiment, access to a thermal infrared camera is the most challenging prerequisite. In the classroom, lack of access is more readily overcome as it serves principally as a visual aid and its absence does not preclude any measurement or calculation. However, the public demonstration largely relies on visuals and interactivity and there are few alternatives that avoid its use. Because visualization, not precision, is the key function of the camera, there is considerable flexibility in choosing an appropriate option. Thermal infrared cameras range from highly technical, precise instruments to simple hand-held or phone-attached cameras (e.g., FLIR One®, Teledyne FLIR,

Wilsonville, OR) that may meet various budgets (as low as ~$100-200 USD, ~ 90-180 €). It is also possible that museums or universities already have thermal infrared cameras that could be borrowed for the land surface activity as it only requires a single, short-duration use.

## 8 Conclusion

    Approaches to ET instruction often hinge solely on theoretical concepts that lack a practical and accessible

framework. In response, we developed a benchtop laboratory activity to address this gap, linking ET to real-world urban heat challenges. Students observe and analyze how different land surfaces and moisture availability impact ET, establishing connections between inherent variations and subsequent changes in heat and water budgets that regulate urban temperatures. Building upon theoretical foundations in ET measurement, students explore urban heat by contrasting representative land surfaces that provide valuable insights into trade-offs among heat, energy, and water cycles. Successfully implemented at three

separate universities and several museum outreach events, the land surface activity demonstrated adaptability to diverse educational and participant requirements. Its hands-on and engaging features offered flexibility in technical depth that could be adjusted to participant background knowledge. Where resources permit, the inclusion of a thermal infrared camera enhances the learning experience by adding a visual dimension, particularly during a campus heat tour. Irrespective of the specific set up, the laboratory activity stimulates critical thinking about key land surface properties and the role of ET processes as drivers

of urban heat. This enables participants to meaningfully discuss effective strategies to mitigate adverse urban heat, drawing upon fundamental relationships explored during the activity.

**Supplemental Materials.** Associated supplemental materials include (a) Instructor Notes to guide a 2.5-3-hour iteration of the laboratory activity and campus heat tour (.pdf), (b) a guide to unit conversions and dimensional analysis for all calculations

using equations within the manuscript (.pdf), and (c) a calculation spreadsheet set up to accept required data for the laboratory activity and automatically calculate latent heat flux, sensible heat flux, and mass and energy balance ET (.xlsx).

**Author Contributions.** Kyle Blount: Conceptualization, Methodology, Formal Analysis, Investigation, Data Curation, Writing – Original Draft, Writing – Review & Editing, and Visualization. Garett Pignotti: Conceptualization, Methodology,

Formal Analysis, Investigation, Data Curation, Writing – Original Draft, Writing – Review & Editing, and Visualization.





Jordyn Wolfand: Conceptualization, Methodology, Formal Analysis, Investigation, Data Curation, Writing – Original Draft, Writing – Review & Editing, and Visualization.

**Competing Interests.** The authors declare that they have no conflict of interest.


**Acknowledgements.** The authors would like to thank the Oregon Museum of Science and Industry (OMSI) for their support in the development of the activity, and we are grateful for the feedback provided by Sean Rooney (OMSI), Jenny Crayne (OMSI), Kevan Moffett (WSU Vancouver), and the OMSI volunteers. Support for this work was provided by NSF CAREER Award #1751377 to Kevan Moffett at Washington State University.

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
