# Peer review of "ET Cool Home: Innovative Educational Activities on Evapotranspiration and Urban Heat"

_Hydrology and Earth System Sciences, 2023_

## Author Response (AR1)

**RESPONSES TO REVIEW COMMENTS**

**MANUSCRIPT HESS-2023-296**

To the editor and reviewers,

We would like to thank you all for your detailed, constructive, and enthusiastic feedback on our manuscript, "ET Cool Home: Innovative Educational Activities on Evapotranspiration and Urban Heat". In accordance with the reviews provided and our responses during the open comment period, we have revised the manuscript. Below, we reproduce the comments provided by the referees and our responses, indented in purple text, to these comments. Where appropriate, we have also reproduced changes to the manuscript, indented in green text, in accordance with these responses. All changes are referenced by line numbers in the revised version of the manuscript. We have also submitted a revised version of the manuscript and a "track change" copy that highlights all revisions made. In addition to the revisions mentioned below, we corrected a few small grammatical errors in the manuscript, which are also reflected in the "track change" copy.

Thank you again for your time and contributions,

Kyle Blount, Garett Pignotti, and Jordy Wolfand, the authors

**REFEREE #1**

Let me start by saying that I wholeheartedly enjoyed reading this manuscript and I applaud the authors for their effort. This manuscript is not a research paper but documents an educational/outreach activity that cuts across both hydrology (ET) and micrometeorology (urban heat). I like the manuscript a lot. One aspect I like in particular is that the authors document all the details, from the theoretical basis, instruments, to lab set-up, to sample question, etc. They even include a calculation spreadsheet that automatically calculate fluxes. Another aspect I like is that the authors always think about the broad context (e.g., understanding coupled natural-human systems and promoting environmental justice).

> We would like to thank you for your time and effort reviewing the manuscript as well as your encouraging and constructive feedback. We are happy to hear that you enjoyed it.

I only have 2 minor comments (they are really just suggestions or clarifying questions). The first is related to the treatment of ground heat flux. The current treatment of ground heat flux (making it zero) is OK but perhaps misses the opportunity to teach/discuss another aspect of urban heat. Urban heat islands are no doubt associated with the lack of

evapotranspiration due to the use of impervious surfaces in cities. Nonetheless, urban heat islands tend to be stronger at night, which is also (perhaps more) related to the fact that these impervious materials tend to store more heat during the day and release them at night (see e.g., Li et al. 2019, Urban heat island: Aerodynamics or imperviousness?.Sci. Adv.5,eaau4299(2019).DOI:10.1126/sciadv.aau4299). This might be why the energy balance approach tends to produce higher ET rates because it did not take into account the heat storage, which is probably not a small term at such time scales for dry surfaces.

> We agree with your assessment of the role of ground heat flux within the activity and manuscript. Though originally developed to focus on ET, the simplifying assumption and subsequent lack of discussion of ground heat flux does miss the opportunity to further and deepen the discussion of urban heat dynamics, particularly regarding the storage and subsequent release of heat from impervious cover. We added this context to the theory section (2.3, lines 129-132, reproduced below) and discussion (7.2, lines 342-344, reproduced below) to better address the impact of this assumption on experimental results within the activity as well as the role of ground heat flux and stored heat in diurnal variations in urban heat island intensity.

> Lines 129-132: "G represents important heat storage characteristics of impervious surfaces, influencing the delayed release of heat at night and the subsequent development of diurnal variations in urban heat island (Li et al., 2019). Despite its significance at the daily scale, G is small over long periods of time (approximately two orders of magnitude smaller than Rn), allowing us to assume that it is equal to zero (Trenberth et al., 2009)."

> Lines 342-344: "Ignoring ground heat flux, for example, may have resulted in relatively higher ET estimates for dry surfaces when using the energy balance approach, as we did not explicitly account for energy lost to storage."

My second comment is related to the temporal change of surface temperature and as a result the temporal change of sensible heat flux. In theory the surface temperature should change throughout during the 90-120 minutes of lab session. Am I right? If the students only take the final surface temperature, does this mean that only the sensible heat flux at the final stage can be calculated? Is this final sensible heat flux used as an approximation for the sensible heat flux during the entire lab session? It would be better to make this clear in one or some of the equations in section 4.2.

> Regarding the second comment, you are correct that surface temperature should change throughout the duration of the activity. We do have students record initial and final surface temperature to facilitate discussion, but the sensible heat flux and subsequent energy balance ET are calculated with the final surface temperature as an approximation for sensible heat flux throughout the duration of the activity. We will add this clarification to section 4.2 (lines 196-197, reproduced below).

Lines 196-197: "Although surface temperature and sensible heat flux vary throughout the activity, sensible heat flux calculation (Eq. 9) uses the final surface temperature as an approximation for sensible heat flux throughout the activity."

Dan Li

**REFEREE #2**

Dear authors, I enjoyed reading your manuscript and find it relevant for future potential application at other UNIs and cities. You have presented an ET educational activity that integrates theory with practice and links ET and UH. Also, you have provided supplementary material which can help future users to implement your proposed method and design of teaching.

Thank you for your review and suggestions - we appreciate the time and constructive feedback you have provided.

I have a few notes for you to address:

1) What are K-12 classrooms? Add explanation in manuscript at appropriate place.

K-12 classrooms are all pre-university classrooms, i.e., primary and secondary school classrooms (students ages 5-17). We have changed "K-12" to "pre-university" in the abstract (line 19) and short summary (line 26).

2) Figure 1 - Add a note that describes the abbreviations in Figure 1.

We have added the definitions of each abbreviation to the figure caption for Figure 1 (lines 71-73, reproduced below).

Lines 71-73: "Abbreviations represent incident shortwave radiation (SWin), outgoing shortwave radiation (SWout), incident longwave radiation (LWin), outgoing longwave radiation (LWout), net radiation (Rn), ground heat flux (G), latent heat flux (LE), and sensible heat flux (Hs)."

3) It would be good to discuss a bit the implications of setting G as 0.

Thank you for this feedback. Referee 1 provided similar comments, and we agree. The simplifying assumption and subsequent lack of discussion of ground heat flux does miss the opportunity to further and deepen the discussion of urban heat dynamics. We added this context to the theory section (2.3, lines 129-132, reproduced below) and discussion (7.2, lines 342-344, reproduced below) to better address the impact of this assumption on experimental results within the activity as

well as the role of ground heat flux and stored heat in diurnal variations in urban heat island intensity.

Lines 129-132: "G represents important heat storage characteristics of impervious surfaces, influencing the delayed release of heat at night and the subsequent development of diurnal variations in urban heat island (Li et al., 2019). Despite its significance at the daily scale, G is small over long periods of time (approximately two orders of magnitude smaller than Rn), allowing us to assume that it is equal to zero (Trenberth et al., 2009)."

Lines 342-344: "Ignoring ground heat flux, for example, may have resulted in relatively higher ET estimates for dry surfaces when using the energy balance approach, as we did not explicitly account for energy lost to storage."

4) In Section 4.1 you mention "ten land surface cores", while in Figure 2 are shown 15 cores. Check this for consistency.

You are correct. Figure 2 has an extra set of land covers (5 additional cores), which were used to spray ethanol and visually demonstrate evaporative cooling with the thermal infrared camera during the demonstration. Have added this clarification to the caption of Figure 2 (lines 171-174, reproduced below).

Lines 171-174: "The first and second sets of cores represent wet and dry conditions. The third set of cores is not represented in this activity but can used to demonstrate instantaneous evaporation by spraying ethanol on the core surface and observing temperature changes associated with ET on the thermal infrared camera display."

5) You used "bare soil", can you add a short info on which type of soil was used?

For the activity, we envisioned soil collection from the local environment for each implementation; however, we will believe that specific soil characterization is outside the scope of the current activity because the variability between land covers will yield much greater differences than changes to soil texture and hydraulic properties. Therefore, the specific soil characterization may prove distracting to readers. However, it is true that the activity could be adapted to test for differences in ET associated with different soils, and we have now noted this as a suggestion for adaptation in section 7.3 (lines 363-364, reproduced below).

Lines 363-364: "Similarly, the activities could be developed to test differences in core properties, such as soil texture, on ET rates."

**REFEREE #3**

The HESS pre-print by Blount et al., summarizes a new hands-on approach to teaching both evapotranspiration and the urban heat island concepts in university courses, while also identifying ways to adapt the activity toward the public. The paper summarizes the theoretical framework of the energy and water balance methods of measuring ET and how they connect to UHI. The authors then summarize their activity, where small cores of different materials common in urban areas (soil with grass, mulch, gravel, etc.) are wetted up, and ET is estimated through both water balance (mass water loss) and energy (temperature change) methods. Students can then compare how different land types and different measurement methods relate to the concept of UHI. It is a great, stand-alone description for anyone teaching ET, going from the theoretical basis all the way to the activity and questions to ask students. It also has great ways to push students to think about historical development of cities, structural racism, and environmental justice. I'm already scrambling to see if I can get my hands on an IR thermometer and heat lamp for a discussion of ET coming up this week in my class.

> Thank you for your detailed, enthusiastic response to the manuscript. We are delighted to hear that you have found value in the material and would love to hear about your experience with your classes if you were able to gain access to an IR thermometer.

I think one potential area for improvement of the paper (and activity) as written is that it focuses so much on difference in land cover impact on ET and linking to UHI and a bit less on leveraging this experiment to help students understand differences in measurement method. To me, this misses a key opportunity to help students more clearly understand the concepts of potential evapotranspiration and actual evapotranspiration, which I have found students at the same level as the authors' (undergraduates) struggle with. I think a bit more explanation of how to use the activity to explain AET and PET would be a great addition, and they could point to then how this activity could to lead into a discussion of the Budyko Curve for anyone who covers that within a course (I realize this last part may be out of scope for many undergraduate courses) or aridity, or water scarcity, or many other things tied to the AET vs. PET balance.

> We appreciate and agree with the comments regarding the opportunity to focus on measurement methods and differentiating between AET and PET, and although we discuss the differences in measurement methods in sample response 1 (lines 243-252), we have aimed to improve these connections in the revisions. Moisture availability and links to AET/PET provide helpful context for many learning environments. We have added discussion of (a) moisture availability and AET/PET in the 'Theoretical Basis' section 2.1 (lines 53-62, reproduced below), (b) discussion of differences in calculated ET based on method in the 'Sample Results' section (lines 245-247, reproduced below), and (c) how the activity can be adapted to focus

more on these consideration in the 'Discussion' section 7.3 (lines 356-361, reproduced below).

Lines 53-62: "ET depends upon both sufficient water supply and adequate energy in the form of solar insolation to occur. In humid environments, the amount of ET that occurs – actual ET – is typically limited by the energy available at the land surface, whereas in more arid regions, actual ET is usually limited by the water available to be evaporated from the land surface. The amount of water that could be evaporated or transpired given unlimited water supply is known as potential ET and constitutes an important parameter for irrigation and water management in arid regions (Dingman, 2015). Calculating potential ET is relatively straightforward based on the incoming radiation at the land surface, though the instrumentation to measure these variables can be expensive. Quantifying actual ET, however, necessitates precise knowledge of each process within a study area, requiring high-quality data that is often more difficult and expensive to measure than for other components of the hydrologic cycle."

Lines 245-247: "This difference indicates differences in actual ET and potential ET in water-limited environments, where ET from wet land surfaces typically exceeds that of dry surfaces."

Lines 356-361: "Although the current iteration of the activity is focused on the relationships between land surface type, moisture, and heat generation, the activity and associated discussion could also focus on the role of water availability on ET. Undergraduate students often have trouble understanding the differences between actual ET and potential ET and between water- and energy-limited environments for ET. A stronger influence could be placed on the role of wet vs. dry land surfaces and lead to discussions of the Budyko Curve, irrigation and reference ET, or water and reservoir management in semi-arid and arid environments (Dingman, 2015)."

The language of the example write ups (section 6.2) is very technical- it's not the kind of language I've seen my students use (for example phrases like "limited moisture retention" and "transpiration potential"). I know students vary in their level of writing and understanding, but were these drawn from student responses? If you have actual student responses, could you summarize some of the phrases from students? I know you may not have had an IRB, which complicates this. But I think two levels of example- that of an expert and that of a student- would be very helpful to set expectations of faculty members/teachers at the correct level.

We agree that there is some confusion about the description of sample responses. What is currently represented is guidance for evaluating responses that includes the content that would be included in an answer that demonstrated mastery of the material (more of an answer key), not sample responses that might be generated by

students. Unfortunately, we do not have an IRB for this study and cannot provide direct student responses from our classes. We have added this clarification prior to the sample responses in lines 233-235, reproduced below.

Lines 233-235: "These sample responses are intended to highlight the content expected in student answers, not replicate responses we have received. At an undergraduate level, students may be expected to understand the underlying processes but often have difficulty articulating these ideas with clarity and correct vocabulary."

Typo I found: line 108 has a comma after "climate change" that seems to me is unnecessary

We have restructured this sentence (now lines 111-112).

1. Does the paper address relevant scientific questions within the scope of HESS?

It is a little unclear to me what type of manuscript type this has been submitted as, but I do believe it clearly falls under "Education and Communication" of a key hydrologic topic within the scope of HESS.

This manuscript was submitted as an "Education and Communication" manuscript.

2. Does the paper present novel concepts, ideas, tools, or data?

Yes, the paper presents a novel approach to teaching ET and linking it to the urban heat island concept. In addition, it could be expanded to more clearly discuss differences between PET and AET, which would be a strong suggestion to the authors.

3. Are substantial conclusions reached?

Not sure how much this fits for an education paper, but yes, they show results from running this activity that show it works.

4. Are the scientific methods and assumptions valid and clearly outlined?

Yes.

5. Are the results sufficient to support the interpretations and conclusions?

Yes

6. Is the description of experiments and calculations sufficiently complete and precise to allow their reproduction by fellow scientists (traceability of results)?

Yes, the set up description is very strong.

7. Do the authors give proper credit to related work and clearly indicate their own new/original contribution?

Yes, the paper does a very good job summarizing the theoretical background behind water and energy approaches to ET, then shows how their activity leverages that theory to teach the concepts in a more 'hands-on' way.

8. Does the title clearly reflect the contents of the paper?

Yes

9. Does the abstract provide a concise and complete summary?

Yes

10. Is the overall presentation well structured and clear?

Yes

11. Is the language fluent and precise?

Yes

12. Are mathematical formulae, symbols, abbreviations, and units correctly defined and used?

Yes

13. Should any parts of the paper (text, formulae, figures, tables) be clarified, reduced, combined, or eliminated?

I'd suggest the caption for Figure 1 should define all variables shown so it can stand alone, if needed.

> Referee #2 also provided feedback to clarify the caption on Figure 1. We have added the definitions of each abbreviation to the figure caption for Figure 1 (lines 71-73, reproduced below).
>
> Lines 71-73: "Abbreviations represent incident shortwave radiation (SWin), outgoing shortwave radiation (SWout), incident longwave radiation (LWin), outgoing longwave radiation (LWout), net radiation (Rn), ground heat flux (G), latent heat flux (LE), and sensible heat flux (Hs)."

14. Are the number and quality of references appropriate?

Yes

15. Is the amount and quality of supplementary material appropriate?

Yes, many thanks for sharing all of the supplemental instructor resources!